# Meniscus-Corrected Method for Broadband Liquid Permittivity Measurements with an Uncalibrated Vector Network Analyzer

**DOI:** 10.3390/s23125401

**Published:** 2023-06-07

**Authors:** Michał Kalisiak, Arkadiusz Lewandowski, Wojciech Wiatr

**Affiliations:** Institute of Electronic Systems, The Faculty of Electronics and Information Technology, Warsaw University of Technology, Nowowiejska 15/19, 00-665 Warsaw, Poland; arkadiusz.lewandowski@pw.edu.pl (A.L.); w.wiatr@elka.pw.edu.pl (W.W.)

**Keywords:** complex permittivity, microwave measurements, vector network analyzer, calibration-independent, coaxial line

## Abstract

We present a novel broadband permittivity characterization method for liquids measured in a semi-open vertically oriented test cell with an uncalibrated vector network analyzer. For this goal, we utilize three scattering matrices measured at different levels of liquid in the cell. With mathematical operations, we remove the effects of systematic measurement errors caused by both the vector network analyzer and a meniscus shaping the top of the liquid samples in such a type of test cell. To the best authors’ knowledge, this is the first of such a calibration-independent method dealing with meniscus. We verify its validity by comparing our results with the data available in the literature and with the outcomes of our previously published calibration-dependent meniscus removal method (MR) for propan-2-ol (IPA), a 50% aqueous solution of IPA and distilled water. The new method yields results comparable with the MR method, at least for IPA and the IPA solution, revealing, however some problems when testing high-loss water samples. Nevertheless, it allows one to cut down on expenditures in the system calibration engaging skilled labor and expensive standards.

## 1. Introduction

Microwave measurement techniques in material research and characterization, being still dynamically developed [1], contribute to the evident progress in the fields of health care, science and technology [2]. All the progress is underlined by new inventions and creative processing of measurement data, which utilizes measurement redundancy and advanced mathematical methods. This processing leads to enhanced characterization accuracy, achieved by eliminating the impact of measuring instruments imperfections, calibration standards, and even the non-ideal shape of measured samples.

There is a vast body of literature on diverse microwave techniques for relative complex permittivity measurement [2,3,4,5]. At microwave frequencies, the most popular and reliable techniques split into broadband and resonant classes [6]. Generally, the resonant ones provide higher accuracy results but usually at discrete frequencies only [7], while the broadband ones are naturally suited for dielectric spectroscopy. Most of the current broadband techniques have their roots in the idea of the transmission–reflection (T/R) method [8], also known as the Nicolson–Ross–Weir (NRW) approach [9,10], which was introduced for characterizing properties of isotropic, homogeneous solid-state materials with a vector network analyzer (VNA). In this classic approach, the VNA needs to be calibrated before the measurement [11,12], and this is usually a tedious task requiring skilled labor and expensive calibration standards. For that reason and because of residual calibration errors that affect the characterization, calibration-independent techniques are thus a highly desirable goal [13,14,15,16,17,18,19,20,21] for replacing the unhandy calibration procedures with mathematical operations performed on measurement results of the uncalibrated VNA.

Although the T/R procedures for measuring liquid and solid materials are generally similar, the solid samples should be carefully processed to fit a waveguide size perfectly; otherwise, serious measurement errors due to air gaps [22] may occur. The air gap problem does not regard liquid samples capable of perfectly fitting space in a cell. However, liquid must be kept in it, e.g., by at least one dielectric plug, set perpendicularly to waveguide walls at the bottom of a semi-open vertically positioned cell [23,24]. The effect of that plug, which, unfortunately, disturbs the measurement, may be eliminated by the appropriate system calibration [23] or mathematical operations, as applied in [24]. Using a semi-open cell provides an opportunity to adjust the volume of a liquid sample pursuant to its attenuation or phase shift introduced. Moreover, it allows for performing a series of measurements at several liquid column heights (cell states), which may yield more information on the liquid tested than a single measurement. However, the permittivity characterization requires knowledge of the sample height and may be interfered with by errors if the upper surface of the liquid sample diverges from a flat and transversal plane due to the meniscus effect or the cell axis being out of the vertical.

The issue of the meniscus has been noticed in [23,24,25,26,27]. To remove the meniscus effect, the authors of [17,28,29] proposed using a holder with two dielectric plugs that determine the volume and the shape of a liquid sample in the cell. However, that solution entails problems with modeling the measurement results, leading to impressive analytical formulae [29,30]. Such boundaries can also be made from very thin, low-loss materials and omitted in the measurements [31]. Filling out a double-plug cell with a liquid might also be troublesome due to air bubbles that need to be utterly removed before the measurement [32]. Moreover, because of a fixed cell size, the usability of such holders is limited to liquids of specific volume and properties, e.g., attenuation. Therefore, using semi-open cells is certainly more advantageous, provided the deformation of the upper surface is accounted for.

Errors in the extraction of the liquid permittivity caused by the upper-surface sample deformations have already been studied by authors who showed in [33] that the errors can be kept small, providing the cell axis is vertically oriented and the meniscus is reproducible at each cell state. For such an instance, we introduced a new approach to broadband permittivity extraction [34], called the meniscus removal (MR) method, since it is capable of removing the meniscus effect using the scattering matrices measured with a calibrated VNA on a semi-open coaxial test cell in just three states. The high accuracy of this method shown in [34] results from the liquid sample column height increment precisely determined from the measured data with the appropriate mathematical calculation. Since it is limited by residual calibration errors, removing the unhandy VNA calibration is becoming the next goal for further development of the MR method.

Pursuing calibration-independent or calibration-free methods for determining the permittivity is an interesting idea propagated in a few papers [16,17,18,19] regarding the technique based on measuring liquids in two-plug (or paraffin film sealing) closed cells whose usability, however, as it was mentioned above, is limited. In contrast to this, such calibration-free measurements performed in semi-open cells at several states provide higher flexibility in the permittivity characterization over broadband frequency ranges. Hasar et al. proposed calibration-independent ideas for a semi-open cell using one [20] and two states [21] of liquid with length measured mechanically, however, not considering the meniscus.

We present in this paper a novel broadband meniscus-corrected permittivity characterization technique for liquids measured in a semi-open vertically oriented test cell with an uncalibrated VNA. Omitting a VNA calibration is the key improvement in comparison with the MR method [34]. To this end, we exploit three scattering matrices measured at different volumes of liquid in the cell. Height increments of the liquid columns, necessary for the calculations, are controlled at each state thanks to a custom liquid dispenser that allows for precisely dosing the liquid. With mathematical operations, we remove the effects of systematic measurement errors caused by both the VNA and an assumed reproducible meniscus shaping the top of each liquid sample in order to determine the complex relative permittivity. Since all the VNA measurements are performed without disassembling the test cell, all the cables and connectors remain intact, and the final results exhibit higher consistency. To the best authors’ knowledge, this is the first such calibration-independent method dealing with the meniscus.

The outline of this paper is as follows. In Section 2, we present the model for a sample measurement and the algorithm for determining the permittivity (with detailed descriptions of solving the final equation in Appendix A). We validate the new method in Section 3 by comparing the results with available reference data and with outcomes of our previously published meniscus removal method [33,34] obtained for propan-2-ol (IPA), a 50% aqueous solution of IPA and distilled water. Then, we summarize the work in the conclusion (Section 4).

## 2. Theory

In this section, we introduce a theory concerning the calibration-independent meniscus-corrected approach for broadband measurements of liquid permittivity in a semi-open, vertically oriented test cell. Figure 1 illustrates a sketch of such a coaxial test fixture used for measuring liquids in three distinct measurement states. These states correspond to three different volumes of the liquid sample within the test cell, denoted by the indices k=1,2,3, respectively.

The mathematical description of this fixture and the theory behind the calibration-independent meniscus-corrected method are established on the essential assumption of single-mode propagation in TEM (transverse electromagnetic) waves within the test fixture. Measurements of such samples are commonly described using the transfer matrices (T), which are associated with the relevant S matrices in the following manner:(1)$$T=\left[\begin{array}{cc} T_{11} & T_{12}\\ T_{21} & T_{22} \end{array}\right]=\frac{1}{S_{21}}\left[\begin{array}{cc} -\det S & S_{11}\\ -S_{22} & 1 \end{array}\right]\;,\;S=\frac{1}{T_{22}}\left[\begin{array}{cc} T_{12} & \det T\\ 1 & -T_{21} \end{array}\right]\;.$$

Each *k*-th measurement in the transfer matrix notation can be described as
(2)$$T_{mk}=E_{A}T_{k}E_{B}\;,$$
where $E_{A}$ is the transfer matrix representing errors at port 1, reflections at the fixture connection and transmission through the part of the empty cell; $E_{B}$ represents the transition from the liquid sample to the dielectric plug (including characteristic impedance change) and then the transmission through the plug, the transition to the air, and the transmission through an airline section and errors at port 2. The “core”’ of the measurements is denoted as $T_{k}$ and is modeled as follows:(3)$$T_{1}=A_{3}A_{2}{\tilde{Q}}_{1}\;,\;T_{2}=A_{3}{\tilde{Q}}_{1}Q_{2}\;,\;T_{3}={\tilde{Q}}_{1}Q_{2}Q_{3}\;.$$

The notation in (Equation 3) corresponds to the sections marked in Figure 1, and so $T_{1}$ represents transmission through an airline section of length $l_{3}$, called $A_{3}$, and then an airline section $A_{2}$ of length $l_{2}$. $A_{n}$, where n=2,3, is described by
(4)$$A_{n}=\left[\begin{array}{cc} a_{n} & 0\\ 0 & a_{n}^{-1} \end{array}\right]\;,\;a_{n}=e^{-\gamma_{a}l_{n}}\;.$$

A transfer matrix modeling the transition from the air to the sample (including characteristic impedance change) with meniscus distortion and transmission through the 1st section of liquid (of length $l_{1}$) is represented by
(5)$${\tilde{Q}}_{1}=\left[\begin{array}{cc} {\tilde{q}}_{11} & {\tilde{q}}_{12}\\ {\tilde{q}}_{21} & {\tilde{q}}_{22} \end{array}\right]\;,$$
where the tilde symbolizes the part of the sample distorted by the meniscus. As ${\tilde{Q}}_{1}$ represents a reciprocal network, its scattering parameters $S_{21}$, $S_{12}$ are equal, and thus, in the transfer matrix notation
(6)$${\tilde{q}}_{22}=\frac{1+{\tilde{q}}_{12}{\tilde{q}}_{21}}{{\tilde{q}}_{11}}\;.$$

Transmission through the symmetrical layer of a liquid sample referring to the characteristic impedance of the line with sample $Q_{n}$ (n=2,3) is expressed as
(7)$$Q_{n}=\left[\begin{array}{cc} q_{n} & 0\\ 0 & q_{n}^{-1} \end{array}\right]\;,\;q_{n}=e^{-\gamma_{q}l_{n}}\;,$$
and thus contains the information about the permittivity we wish to extract.

By multiplying the transfer matrix for the first state by the inverse of the *n*-th state (n=2,3), we eliminate the unknown $E_{B}$ and obtain a “ratio” matrix
(8)$$R_{m\;n,1}=T_{m1}T_{mn}^{-1}=E_{A}T_{1}T_{n}^{-1}E_{A}^{-1}=E_{A}R_{n,1}E_{A}^{-1}\;.$$

Since (Equation 8) represents a matrix similarity transformation, the trace operation also eliminates $E_{A}$ from the formula
(9)$$\mathrm{tr}\;R_{m\;n,1}=\mathrm{tr}\;R_{n,1}\;.$$

By inserting (Equation 4)–(Equation 7) into (Equation 9), for first two states, we obtain
(10)$$\mathrm{tr}\;R_{m\;2,1}=\tilde{q}\left(\frac{a_{2}}{q_{2}}+\frac{q_{2}}{a_{2}}-a_{2}q_{2}-\frac{1}{a_{2}q_{2}}\right)+\frac{a_{2}}{q_{2}}+\frac{q_{2}}{a_{2}}\;,$$
where $\tilde{q}={\tilde{q}}_{12}{\tilde{q}}_{21}$ is the only remaining variable directly related to the sample part distorted by the meniscus, which can be eliminated with the data measured for another state. From (Equation 10), we obtain
(11)q˜=−trRm2,1−2coshγa−γq1l24sinhγal2sinhγql2.

Similarly for the pair of states 1 and 3:(12)q˜=−trRm3,1−2coshγa−γq1l234sinhγal23sinhγql23,
where $l_{23}=l_{2}+l_{3}$. Equating (Equation 11) and (Equation 12) leads to the following equation, whose left side we denote as Ψ,
(13)Ψ=trRm2,1−2coshγa−γq1l2sinhγal2sinhγql2−trRm3,1−2coshγa−γq1l23sinhγal23sinhγql23=0,
which consists of two real, frequency-independent values, $l_{2}$ and $l_{3}$, and one complex frequency-dependent quantity:(14)$$\gamma_{q}=\alpha_{q}+j\beta_{q},$$
where $\alpha_{q}$ is an attenuation constant and $\beta_{q}$ is a phase constant of the line with the sample.

The increments in the height of the liquid columns $l_{2}$ and $l_{3}$ can be tracked thanks to a custom liquid dispenser, described in Section 3.1. Therefore, in (Equation 13), only one complex frequency-dependent $\gamma_{q}$ remains unknown; thus, $\Psi =\Psi (f,\;\gamma_{q})$. To solve (Equation 13), we use the fsolve algorithm from MATLAB, transforming Ψ to the system of two real equations with two real unknowns: $\alpha_{q}$ and $\beta_{q}$.
(15)Fαq,βq=ReΨImΨ.

We solve (Equation 15) individually at each frequency. Since (Equation 13) has an infinite number of solutions, we devised a special solution strategy based on multiple starts of the fsolve algorithm from different starting points and selected the solutions so as to provide the continuity in $\gamma_{q}$ frequency dependence. A detailed description of the algorithm is presented in Appendix A.

For a nonmagnetic sample ($\mu_{r}=1$), the complex relative permittivity of the sample [35] is
(16)$$\varepsilon_{q}=\varepsilon_{q}^{\prime }-j\varepsilon_{q}^{\prime \prime }=-{\left(\gamma_{q}\frac{c}{\omega }\right)}^{2}\;,$$
where *c* is the speed of light in a vacuum and ω=2πf is the angular frequency.

## 3. Experimental Results

Our coaxial test fixture design for the liquid permittivity measurements was inspired by the concept outlined in [23]. The fixture utilized a 7 mm line standard with laboratory precision connectors (LPC-7) [36]. Its body was machined from a brass rod in an in-house workshop and equipped with two small holes in the wall: first, at the bottom for dosing liquids and, second, at the top, to let excess air out. Along with a center conductor and a PTFE annular plug, as illustrated in Figure 2, it was put together and vertically mounted in the measurement setup presented in Figure 3.

Measurements of the scattering parameters were performed with the uncalibrated VNA Rohde & Schwarz ZVA50 in the 0.1–18 GHz frequency range. VNA’s switching errors, resulting from changes in impedance and transmission when switching the direction of forward and reverse excitation, were eliminated from all the two-port measurements using the technique proposed in [37].

We measured the fixture in three states of the liquid, carefully dosed with a liquid dispenser visible in Figure 3, by counting the integer number of full knob rotations. Then, we applied the calibration-independent meniscus-corrected algorithm (CI).

For comparing the results of the new CI method and the earlier MR one [34], which requires the calibrated VNA, we corrected the S-matrices using the calibration terms determined in the course of the TRM (thru–reflect–match) calibration and the relevant algorithms known from [38,39]. For that reason, we performed additional measurements of thru, short and match standards, as well as the empty cell needed in the MR method. Because this technique requires only 2 levels of liquid, we calculated three values of permittivity for pairs $\left(1,2\right)$, $\left(1,3\right)$, $\left(2,3\right)$, and their mean value treated as the final result of the MR method.

The height determination of the liquid columns is analyzed below, in Section 3.1. The results of the permittivity obtained for the calibration-independent meniscus-corrected technique for IPA, a 50% aqueous solution of IPA and distilled water are presented and discussed in Section 3.2, Section 3.3 and Section 3.4, respectively.

### 3.1. Determination of the Column Height
Increment

The liquid dosing utility determines the sample height increments $l_{2}$ and $l_{3}$. It uses a screw mechanism connected with the piston of the 1mL precision syringe, as shown in Figure 3. The height *l* of a liquid column of volume *V* in coaxial line with outer and inner conductor diameters *D* and *d*, respectively, is
(17)$$l=\frac{4V}{\pi \left(D^{2}-d^{2}\right)}\;.$$
The liquid column height increment per *n* full rotation of the knob for the 7 mm coaxial line (D=7mm, d=3.04mm) is estimated as
(18)l=376n±4n+71μm,
where 4nμm is the error due to the syringe scale resolution and 7μm is the error resulting from uncertainty of the knob-mark setting.

In Table 1, we collate the height increments (marked in Figure 1) determined from the full number of knob rotations along with their uncertainties. The number of rotations was chosen pursuant to the attenuation of the measured liquids, and thus for highly attenuating water, the increments are smaller than those for IPA. The height increments may be compared with electrical measurements of the length available in the MR method [34]. Since the deviations given in the bottom row of Table 1 are much smaller than the estimated uncertainty, the consistency of lengths $l_{2}$, $l_{3}$ determined from mechanical and electrical measurements is confirmed.

### 3.2. Propan-2-ol

The results of the IPA permittivity measurement obtained with the new CI method are shown in Figure 4. As a reference, we used the results obtained with the MR method [34]. We also show the available literature data from NPL Report [40], however, certified only below 5 GHz. Our measurements were performed at the ambient temperature 24 ${}^{{}^{\circ}}$C.

We can observe good consistency between the calibration-independent and calibration-dependent MR methods’ outcomes in almost the entire measuring range. Those results confirm the robustness of the new method which does not require expensive standards for the VNA calibration. The measured characteristics have a similar shape to the NPL data [40], although they are shifted in frequency. According to the model from [40], this may be caused by a high sensitivity of the IPA relaxation frequency to the temperature.

The permittivity results below 1GHz differ much from the MR outcome. This is probably because the CI method strongly relies on transmission, which does not provide valuable information at low frequencies due to the minor phase changes. To verify this hypothesis, we performed the measurements for shorter and longer length increments $l_{2}$, $l_{3}$, and the results for frequencies up to 2 GHz are visible in Figure 5. The conclusion from the observation is that with longer segments and a higher phase change in the sample, the permittivity results stay valid for a lower frequency, which confirms the above hypothesis.

### 3.3. 50% Aqueous Solution of IPA

The 50% aqueous solution of IPA was prepared with the definition of the volume fraction [41]. Therefore, the volume of IPA divided by the sum of volumes of water and IPA prior to mixing $\frac{V_{IPA}}{V_{IPA}+V_{H_{2}O}}$ is equal 0.5. The permittivity results calculated for this solution are shown in Figure 6. Like the results discussed in Section 3.2, we used the mean value of the results obtained with the MR method [34] as the reference. Unfortunately, we have not obtained relevant literature data for comparison, except just static permittivity at room temperature [42] shown with green crosses, confirming that values at the low-frequency end are reasonable. The measurements were performed at 25 ${}^{{}^{\circ}}$C.

The permittivity of the IPA solution obtained with the calibration-independent method demonstrates very good consistency with the outcome of the MR method in almost the entire measuring range, except at the low-frequency end. Importantly, at high frequencies, the results obtained with the MR method exhibit small ripples, characteristic for the residual calibration errors arising from calibration standard imperfections, while the permittivity calculated with the new method yields smoother curves, as exposed in Figure 6c with a zoomed part of $\varepsilon_{q}^{\prime \prime }$ in the range 10–14 GHz.

### 3.4. Distilled Water

Measuring distilled water in this frequency range is a challenging task due to its high attenuation. From that point of view, the first level $l_{1}$ and the increments $l_{2}$ and $l_{3}$ should be as short as possible. On the other hand, as discussed in Section 3.2, short sections cause errors at low frequencies due to small phase changes introduced by the samples. The permittivity results of water are shown in Figure 7. As a reference, we used the data from [43]. The results obtained with the MR method [34] are also presented. The measurements were performed at 23 ${}^{{}^{\circ}}$C.

Despite very thin layers of sections $l_{1}$, $l_{2}$, and $l_{3}$, obtaining the meaningful results with the CI method in the high-frequency range was impossible. Furthermore, high errors at the low-frequency end reach up to about 5 GHz. The MR method yields better outcomes at low as well as high frequencies. The probable reason for that is already mentioned to be a high attenuation of water.

In Figure 8, we can observe the raw $\left|S_{21}\right|$ (dB) parameter measured for all three liquids and states (k=1,2,3; Figure 1). For water, all the states show weak transmission at the high-frequency end, which may cause errors in the permittivity determination.

## 4. Discussion and Conclusions

We presented a novel method for liquid broadband permittivity characterization in semi-open vertically oriented test cells with an uncalibrated two-port VNA. This is, to the authors’ best knowledge, the first step towards the development of calibration-independent methods dealing with menisci shaping the top of the liquid in such a type of test cell. This novel method exploits three levels of liquid measured to eliminate the effects of the VNA systematic errors and a quantity caused by a reproducible meniscus. The increments in the liquid column height between the states are precisely set thanks to a custom liquid dispenser that allows for dosing the liquid volume without disassembling the test cell. Since all the cables and connectors remain intact, the measurement results exhibit higher consistency.

We validated the new method by comparing the obtained results with available literature data and with the outcomes of our previously published calibration-dependent meniscus removal method (MR) for propan-2-ol (IPA), a 50% aqueous solution of IPA and distilled water. The new method yields results comparable with the MR method, at least for IPA and the solution, and does not require expensive standards for the VNA calibration.

This method has a low-frequency cutoff that arises from too-low phase changes inside of the liquid under testing, which was confirmed using the experiments with different volumes of IPA (Figure 5). Unfortunately, the new algorithm also does not cope well with high-loss liquids, such as water. Poor outcomes at high frequencies result from transmission errors at low $\left|S_{21}\right|$ levels. Future research on calibration-independent methods could focus on obtaining the length of the samples with electrical techniques.

## Figures and Tables

**Figure 1 sensors-23-05401-f001:**
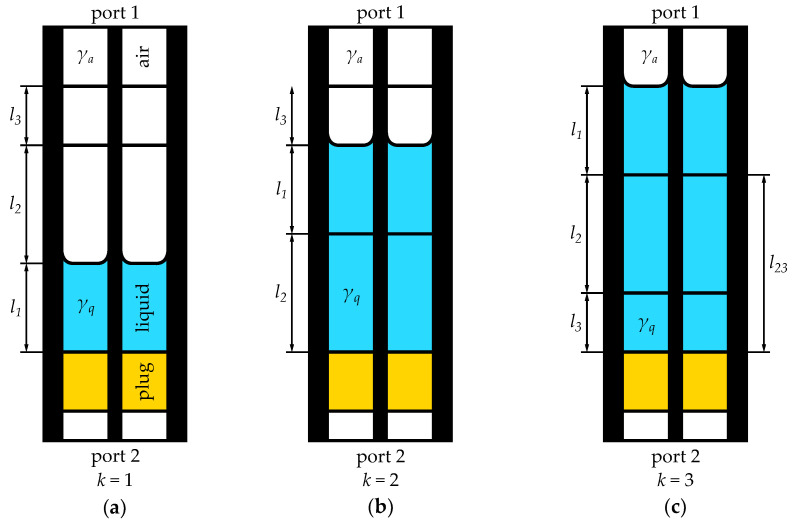
Sketch of the semi-open coaxial fixture in its three measurement states (k=1,2,3) of liquid under test distorted by the meniscus: (**a**–**c**). List of symbols: $l_{i}$—the length of *i*-th section (i=1,2,3); $\gamma_{a}$ and $\gamma_{q}$—propagation constants for the air and liquid sample, respectively. Colour blue represents the liquid under test, yellow—the plug, white—the air.

**Figure 2 sensors-23-05401-f002:**
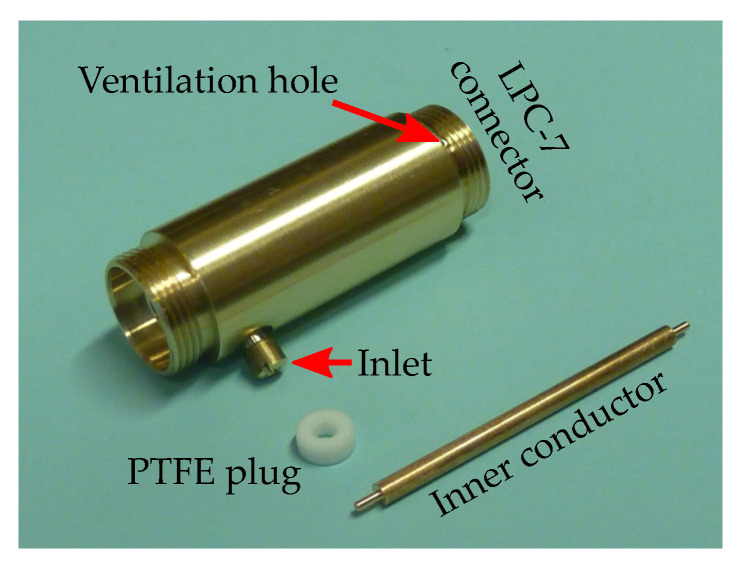
Disassembled fixture for measuring liquids.

**Figure 3 sensors-23-05401-f003:**
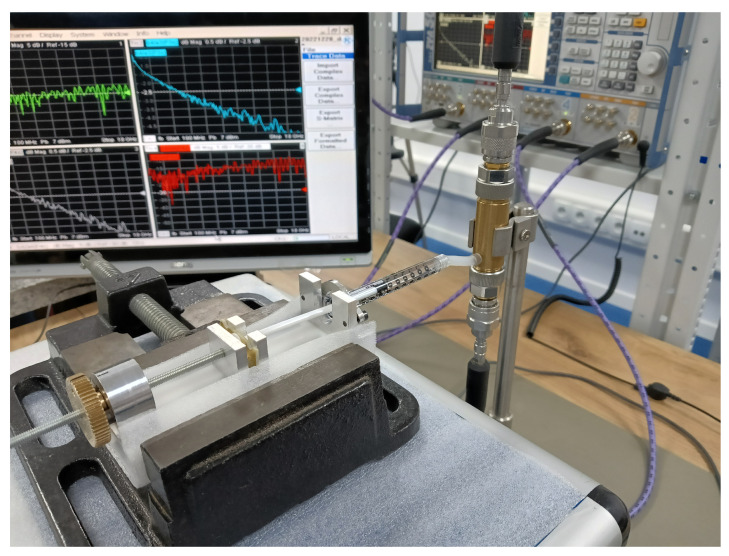
Measurement setup: the semi-open coaxial fixture in 7 mm standard connected to the VNA with the liquid dosing utility.

**Figure 4 sensors-23-05401-f004:**
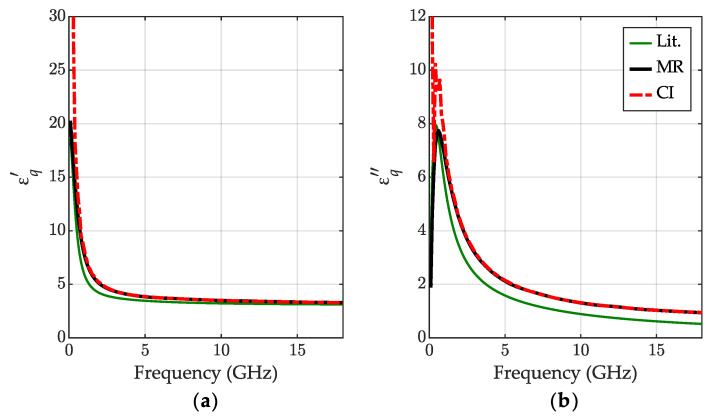
The relative permittivity for IPA at 24 ${}^{{}^{\circ}}$C: (**a**) the real part $\varepsilon_{q}^{\prime }$ and (**b**) the imaginary part $\varepsilon_{q}^{\prime \prime }$ obtained for the calibration-independent meniscus-corrected method (CI)—red dashed-dotted lines and the meniscus removal method (MR)—black lines; the literature data [40]—green lines.

**Figure 5 sensors-23-05401-f005:**
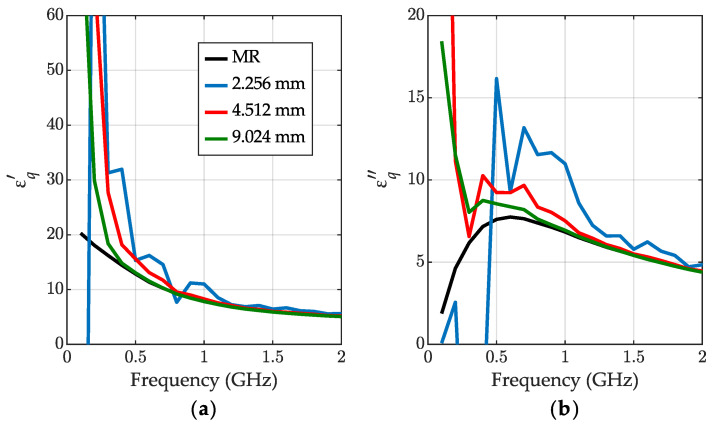
The relative permittivity for IPA at the low-frequency end: (**a**) the real part $\varepsilon_{q}^{\prime }$ and (**b**) the imaginary part $\varepsilon_{q}^{\prime \prime }$ obtained for the calibration-independent meniscus-corrected method (CI) for different increments $l_{2}=l_{3}$: 2.256 mm—blue dashed-dotted lines, 4.512 mm—red dashed-dotted lines, 9.024 mm—green dashed-dotted lines. The MR method results for 4.512 mm—black lines.

**Figure 6 sensors-23-05401-f006:**
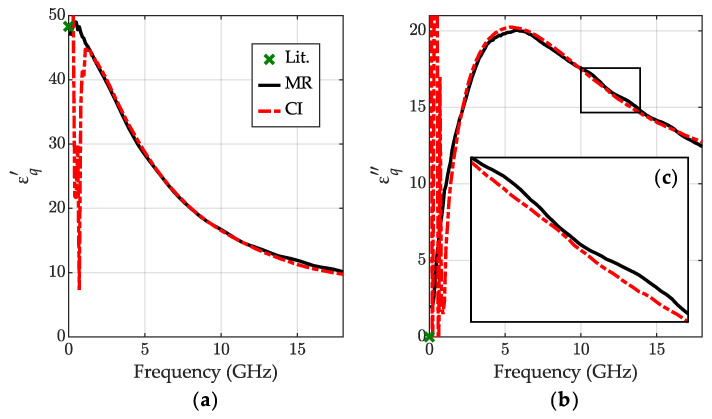
The relative permittivity for 50% (the volume fraction) aqueous solution of IPA at 25 ${}^{{}^{\circ}}$C: (**a**) the real part $\varepsilon_{q}^{\prime }$ and (**b**) the imaginary part $\varepsilon_{q}^{\prime \prime }$ with (**c**) zoomed part for 10–14 GHz obtained for the calibration-independent meniscus-corrected method (CI)—red dashed-dotted lines and the meniscus removal method (MR)—black lines; the literature data (static permittivity) [42]—green cross.

**Figure 7 sensors-23-05401-f007:**
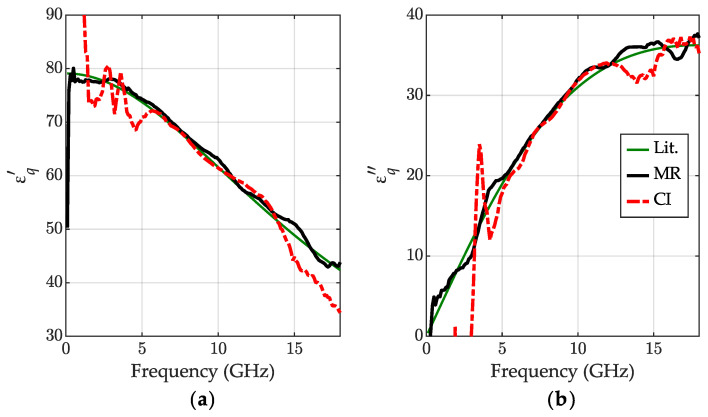
The relative permittivity for distilled water at 23 ${}^{{}^{\circ}}$C: (**a**) the real part $\varepsilon_{q}^{\prime }$ and (**b**) the imaginary part $\varepsilon_{q}^{\prime \prime }$ obtained for the calibration-independent meniscus-corrected method (CI)—red dashed-dotted lines and the meniscus removal method (MR)—black lines; the literature data [43]—green lines.

**Figure 8 sensors-23-05401-f008:**
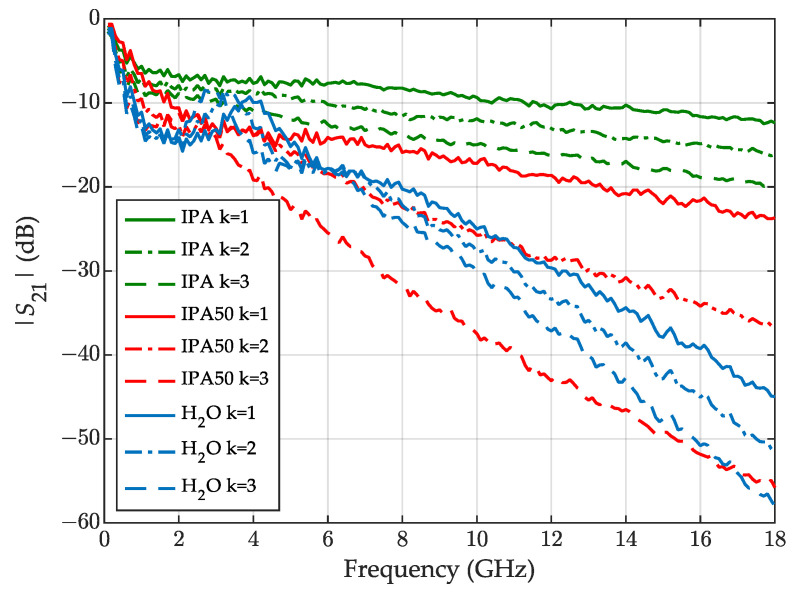
The raw $\left|S_{21}\right|$ (dB) parameter measured for IPA—green lines, a 50% aqueous solution of IPA—red lines, distilled water—blue lines, for three levels of liquid k=1—solid lines, k=2—dashed-dotted lines, k=3—dashed lines.

**Table 1 sensors-23-05401-t001:** Lengths of the sections $l_{2}$ and $l_{3}$ (see Figure 1) for IPA, a 50% aqueous solution of IPA and distilled water, determined with the liquid dosing utility and confirmed electrically with the MR method [34].

	IPA		IPA50		H_2_O	
	$l_{2}$	$l_{3}$	$l_{2}$	$l_{3}$	$l_{2}$	$l_{3}$
Number of knob rotations	12	12	6	9	2	2
Length *l* $\left(\mu m\right)$	4512	4512	2256	3384	752	752
Uncertainty $\left(\mu m\right)$	55	55	31	43	15	15
Length MR $l_{MR}$ $\left(\mu m\right)$	4528	4531	2253	3383	747	749
$\left|l-l_{MR}\right|\;$ $\left(\mu m\right)$	16	19	3	1	5	3

## Data Availability

The data presented in this paper are available from the corresponding author on request.

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
