# Peer review of "Meniscus-Corrected Method for Broadband Liquid Permittivity Measurements with an Uncalibrated Vector Network Analyzer"

_sensors, 2023, doi:10.3390/s23125401_

Round 1

Reviewer 1 Report

An interesting paper describing a follow-on for the authors' work on meniscus correction in permittivity measurements using a semi-open coaxial probe. A few comments on the article below:

A good introduction however, the referencing is excessive. There are several instances where a large number of references are reported without any analysis of the work. For instance, the literature on complex permittivity [2-9], NRW approach [12-16], and need for calibration [19-31]. References should be chosen selectively.

Some clarification necessary over how this work departs from the previous work. Is it just the ability to use an uncalibrated VNA?

Description of the text fixture is very brief. Please provide the full details, sufficient at least for other researchers to replicate this work.

Is temperature of the solution considered/measured? Seems like an important variable, so if not, why is it not accounted for.

Figure 3 and similar - it is difficult to see the significant effects in these graphs. Perhaps a semilogx plot would be better?

The language is generally good with just minor proof-read required.

Author Response

Dear Reviewer, we would like to thank you for your detailed review, which has helped us to improve the paper, and for your kind words.

Point 1: A good introduction however, the referencing is excessive. There are several instances where a large number of references are reported without any analysis of the work. For instance, the literature on complex permittivity [2-9], NRW approach [12-16], and need for calibration [19-31]. References should be chosen selectively.

Response 1: Thank you for this suggestion. We have limited the number of references, leaving 4 regarding microwave techniques for permittivity measurement, 2 for NRW, 9 for calibration-independent methods: 3 regarding measurements of solid materials, and 6 for liquid materials, analyzed in the introduction, paragraph 6.

Point 2: Some clarification necessary over how this work departs from the previous work. Is it just the ability to use an uncalibrated VNA?

Response 2: Yes. To clarify the improvement of this method, we have added a sentence “Omitting a VNA calibration is the key improvement in comparison with the MR method” in the introduction, paragraph 7.

Point 3: Description of the text fixture is very brief. Please provide the full details, sufficient at least for other researchers to replicate this work.

Response 3: Upon the Reviewer's comment, we have added a picture of the disassembled fixture to complete the description – Figure 2.

Point 4: Is temperature of the solution considered/measured? Seems like an important variable, so if not, why is it not accounted for.

Response 4: The temperature is important, indeed, and was measured for every liquid and is clarified in the description in the corresponding figures with permittivity results. For the reader's convenience, we have added this information also in the main text.

Point 5: Figure 3 and similar - it is difficult to see the significant effects in these graphs. Perhaps a semilogx plot would be better?

Response 5: Thank you for this comment. This is an important observation that probably should be highlighted more in the text. Indistinguishable results for calibration-dependent and -independent methods are excellent news as it means that no expensive standards are needed to obtain the same results for IPA, or even better in the case of a 50% solution. However, high-permittivity water is an example where the novel method does not perform well, in contrast to the calibration-dependent MR method.

We have added additional descriptions of the results to emphasize this information and Figure 6c with the zoom of the exciting part of plot 6b.

The low-frequency results are poor (which is extensively discussed in the text). The interesting results cover one decade of the frequency; thus, linear frequency is a better solution than semilogx.

Point 6: The language is generally good with just minor proof-read required.

Response 6: We have performed additional corrections of the language (all marked in the text).

Thank you again for your review and the comments that have helped us to improve the paper.

Reviewer 2 Report

Dear Editor:

Thank you for giving me this opportunity to review the manuscript. English language is fine and introduction is well written in this manuscript.

Question: The title can be changed to highlight the topic of the paper. That is the subject should be clearer and more specific.

Author Response

Dear Reviewer, we would like to thank you for your review, and for your kind words.

Point 1: The title can be changed to highlight the topic of the paper. That is the subject should be clearer and more specific.

Response 1: Thank you for this comment. We have changed
"Calibration-Independent Meniscus Corrected Method for Broadband Liquid Permittivity Measurements"
to:
"Meniscus Corrected Method for Broadband Liquid Permittivity Measurements with an Uncalibrated Vector Network Analyzer"

We have also added a keyword "calibration-independent."

Thank you again for your review and the comment that have helped us to improve the paper.

Reviewer 3 Report

The manuscript presents a novel permittivity characterization method for liquids by removing effects of systematic measurement errors caused by both the vector network analyzer and a meniscus shaping with mathematical operations. It is recommended for publication after addressing these issues.

1.Why choose 24°C in Fig.3? It is recommended to add other temperatures for comparison.

2.Why does the permittivity in Fig.4a and 4b appear negative? Is the instrument incorrectly calibrated?

3.This may be caused by a high sensitivity of the IPA relaxation frequency to the temperature.” is mentioned in the manuscript, but there is no relevant temperature change test to verify.

4. There are some issues in some figures. For example ,the labels in Fig.A2, A3 and A4 block the curves in the figure.

5. The authors should assess the advantages and disadvantages of this method compared to other similar methods.

Author Response

Dear Reviewer, we would like to thank you for your detailed review, which has helped us to improve the paper.

Point 1: Why choose 24°C in Fig.3? It is recommended to add other temperatures for comparison.

Response 1: That was the ambient temperature measured in the laboratory during the measurement. The scope of this work is not the characterization of liquids in the function of temperature but to provide a novel method of permittivity measurement.

Point 2: Why does the permittivity in Fig.4a and 4b appear negative? Is the instrument incorrectly calibrated?

Response 2: The VNA is not calibrated for the method presented in this paper. The permittivity errors (including the negative values) at the low-frequency end result from too small phase change of measured signals. This topic is a serious issue in this method, as it strongly relies on the transmission coefficient, and is illustrated by this Figure (now 5a, b).

Point 3: This may be caused by a high sensitivity of the IPA relaxation frequency to the temperature.” is mentioned in the manuscript, but there is no relevant temperature change test to verify.

Response 3: Thank you for this comment. We have corrected this part which could lead to ambiguity. Please find below the explanation.

The cited sentence is a hypothesis trying to explain the difference in the results in measurements (for both the new method and our previous MR method) and the model from the NPL report.

The sensitivity of IPA relaxation frequency to the temperature is not the conclusion from our measurements but the observation resulting from the model introduced in the NPL report. We added the reference to clarify this statement.

As we mentioned before, the characterization of liquids in the function of temperature is out of the scope of this work.

We hope this explanation and the corrections we have made are sufficient.

Point 4: There are some issues in some figures. For example ,the labels in Fig.A2, A3 and A4 block the curves in the figure.

Response 4: Thank you, we have corrected the Figures.

Point 5: The authors should assess the advantages and disadvantages of this method compared to other similar methods.

Response 5: Thank you for this comment. However, we think this aspect is covered in the paper. Let us summarize it briefly. The advantages and disadvantages of the novel method are highlighted in the summary.

Main advantages of the novel method:

  • using the semi-open cell (with one plug), which is more versatile than two-plug cells, with the elimination of the meniscus from the measurements
  • no need for VNA calibration.

The main disadvantages regard:

  • errors at low frequencies,
  • errors in measurements of liquids with high losses.

The introduction also covers the state of the art regarding liquid measurements. Paragraph 2 discusses different types of measurements performed at microwave frequencies. Paragraph 3 explains different approaches using two-plug cells (closed) and semi-open ones. 6th paragraph discusses the state of the art regarding calibration-independent measurements of liquid permittivity with the pros and cons of two main approaches (closed and semi-open cells). One of the disadvantages of measurements in semi-open cells is the meniscus shaping the top surface of the liquid tested (paragraph 4). Examples of measurements in semi-open cells are cited [20,21] (the new numeration).

Nevertheless, no techniques – to the authors' best knowledge – would deal with the meniscus problem in semi-open cells (7th paragraph). The two main advantages of this novel method are eliminating the meniscus from the measurements using the uncalibrated VNA.

Thank you again for your review and the comments that have helped us to improve the paper.

Reviewer 4 Report

Just a minor comment only. The author's need to to provide details on the temperatures at which the calibration and measurements were performed.

Proof reading is recommended

Author Response

Dear Reviewer, we would like to thank you for your review, which has helped us to improve the paper.

Point 1: The author's need to to provide details on the temperatures at which the calibration and measurements were performed.

Response 1: Thank you for this comment. The temperature was clarified in the description in corresponding figures with permittivity results. For the reader's convenience, we have added this information also in the main text.

Point 2: Proof reading is recommended

Response 2: We have performed additional corrections of the language (all marked in the text).

Thank you again for your review and the comments that have helped us to improve the paper.